# A Co-Created Tool to Help Counter Health Misinformation for Spanish-Speaking Communities in the San Francisco Bay Area

**DOI:** 10.3390/ijerph21030294

**Published:** 2024-03-02

**Authors:** Lucía Abascal Miguel, Andres Maiorana, Gustavo Santa Roza Saggese, Chadwick K. Campbell, Beth Bourdeau, Emily A. Arnold

**Affiliations:** 1Institute for Global Health Sciences, University of California San Francisco, San Francisco, San Francisco, CA 94158, USA; 2Division of Prevention Science, University of California San Francisco, San Francisco, CA 94158, USA; andres.maiorana@ucsf.edu (A.M.); beth.bourdeau@ucsf.edu (B.B.); emily.arnold@ucsf.edu (E.A.A.); 3Santa Casa School of Medical Sciences, São Paulo 01221-020, Brazil; gsrsaggese@gmail.com; 4Herbert Wertheim School of Public Health & Human Longevity Science, University of California, San Diego, CA 92093, USA; ckc003@health.ucsd.edu

**Keywords:** health misinformation, Spanish-speaking communities, COVID-19 vaccine, digital health literacy, community-based organization (CBO), co-design workshops

## Abstract

Background: Health misinformation, which was particularly prevalent during the COVID-19 pandemic, hampers public health initiatives. Spanish-speaking communities in the San Francisco Bay Area may be especially affected due to low digital health literacy and skepticism towards science and healthcare experts. Our study aims to develop a checklist to counter misinformation, grounded in community insights. Methods: We adopted a multistage approach to understanding barriers to COVID-19 vaccine uptake in Spanish-speaking populations in Alameda and San Francisco counties. Initial work included key informant and community interviews. Partnering with a community-based organization (CBO), we organized co-design workshops in July 2022 to develop a practical tool for identifying misinformation. Template analysis identified key themes for actionable steps, such as source evaluation and content assessment. From this, we developed a Spanish-language checklist. Findings: During formative interviews, misinformation was identified as a major obstacle to vaccine uptake. Three co-design workshops with 15 Spanish-speaking women resulted in a 10-step checklist for tackling health misinformation. Participants highlighted the need for scrutinizing sources and assessing messenger credibility, and cues in visual content that could instill fear. The checklist offers a pragmatic approach to source verification and information assessment, supplemented by resources from local CBOs. Conclusion: We have co-created a targeted checklist for Spanish-speaking communities to identify and counter health misinformation. Such specialized tools are essential for populations that are more susceptible to misinformation, enabling them to differentiate between credible and non-credible information.

## 1. Background

The COVID-19 pandemic has exposed significant health disparities among racial and ethnic groups in the United States [1]. Latino individuals in California were 8.1 times more likely to reside in households with a higher risk of exposure, were more likely to have severe outcomes at the outset of the pandemic, and are currently less likely to have received vaccinations compared to their white counterparts [2]. These disparities are influenced by social, structural, and individual factors, with health information and education also playing a role [3].

Many people used social media during the pandemic to find information about COVID-19. However, due to the increase in health-related content circulating on social media that needs more proper scrutiny and fact-checking, this increase in online information has led to an “infodemic” of misinformation and false claims [4,5,6]. Studies have shown that many posts about COVID-19 on social media are untrustworthy and contain false information and conspiracy theories about the disease and vaccines [5,6]. This surge in health-related content on social media has made it challenging to distinguish accurate information from falsehoods [7]. False information tends to spread faster and farther than accurate information [8,9].

Digital health literacy involves the ability to access health information online and understand and apply it accurately. Low levels of digital health literacy contribute to the spread of COVID-19-related online misinformation [10,11]. Latinos and Spanish-speaking individuals are particularly vulnerable to misinformation. Spanish-speaking households rely more on social media for health information and are more likely to consume and share misinformation online than the general population [12,13].

Social media platforms, such as Facebook, are less effective at identifying and flagging Spanish-language than English-language misinformation, further exacerbating its spread. One study found that the platform failed to flag 70% of misinformation in Spanish compared to 29% in English [14]. Moreover, most social media platforms invest about nine times less in fact-checking in languages other than English, further amplifying the risk of spreading misinformation [15]. This vulnerability is exacerbated by a history of discrimination, medical racism, and limited access to healthcare, which has created a foundation of mistrust that allows Spanish-language COVID-19 vaccine misinformation to thrive on social media platforms [12].

Participatory design methods are increasingly recognized as a valuable approach for creating health and public health interventions [16]. Participatory design methods lead to more relevant, effective, and sustainable public health solutions by actively involving end users, fostering collaboration, and promoting adaptability. Within participatory design, co-design workshops enable users and researchers to exchange and develop ideas, aiming to ensure that the tools being created are rooted in users’ lived experiences, while actively involving them in the design process [17]. User narratives, such as stories and scenarios, may also be employed in co-design to communicate design concepts and envision their potential applications [18].

Although originally developed and primarily used for new technologies and mHealth, these approaches can be adapted for developing more traditional and non-technological health tools, including information and education campaigns, infographics, and more. The collaborative development process provides vital insights into how end users interact with health tools, leading to relevant and timely solutions for health issues. As both tools and the sociocultural context of end users continuously evolve, the collaborative development process should remain dynamic [19].

Given the prolific nature of misinformation and its impact on the health-related decisions of Spanish-speakers, efficient strategies and tools are needed to help identify misinformation on the internet. This study aims to explore the content, causes, and sources of misinformation affecting Spanish speakers in the San Francisco Bay Area to co-design a user-friendly checklist for identifying and countering online misinformation about COVID-19 vaccination.

## 2. Methods

We employed a two-stage methodology, through formative interviews and then through community workshops, to understand and address barriers to COVID-19 vaccine uptake in Spanish-speaking populations.

## 3. Formative Interviews and Groups

We conducted interviews and group interviews with key informants (KIs) and community members in San Francisco and Alameda Counties, California from August to December 2021. KIs included healthcare professionals, community-based organization (CBO) personnel, county health department members, and community leaders. Community members were recruited through social media and CBOs. Interviews were conducted via Zoom, transcribed, and analyzed using template analysis. Topics included community perceptions of COVID-19, vaccine barriers and facilitators, misinformation, and intervention recommendations.

As we analyzed the formative phase data, we uncovered misinformation as a recurring theme and identified it as a significant barrier to vaccine uptake among the Spanish-speaking community. In response to this finding, we partnered with a CBO representing Spanish-speaking members and organized co-design workshops in July 2022. Through these workshops, we aimed to explore the problem of misinformation further and to collaboratively develop a practical tool to identify and counteract false information.

## 4. Workshops

### 4.1. Data Collection

We collaborated with a local CBO, Mujeres Activas y Unidas (MUA), to recruit participants for three 1-h workshops. Mujeres Unidas y Activas (MUA) is an organization comprised of Latina and Indigenous immigrant women in the San Francisco Bay Area. It is dedicated to empowering both individual and community strengths with the aim of achieving social and economic justice. Interested participants were contacted by a Spanish-speaking member of the research team who explained this study to them. Workshops were conducted in Spanish by a bilingual moderator and note-taker over Zoom and were audio-recorded and transcribed. The workshops began with an open-ended discussion where participants shared their experiences assessing the veracity of COVID-19 vaccine-related information. Participants discussed the source, messenger, and content of the information and shared strategies or techniques they used to identify misinformation or disinformation.

Then, participants engaged in a practical exercise comparing two pieces of information on COVID-19 boosters. Both posts were shared side by side without revealing which information was true or false. One piece of information was from Dr. Mercola’s Facebook site in Spanish, presenting a hand in a blue glove holding a syringe with a conspicuously long needle, alongside the question, “Why do people with all of their booster shots continue to get COVID and the unvaccinated don’t?” This was coupled with a link suggesting a grave health risk from booster shots. In stark contrast, the CDC’s Spanish Facebook post, verified as well, displayed a straightforward cartoon of a contented man with a band-aid on his arm, promoting the message that booster shots can enhance or reinstate waning protection against COVID-19. Both accounts, bearing the blue checkmark of Facebook verification, were posted during the same week in May 2022.

Participants were asked to compare the two pieces of information and identify any differences that could help distinguish between accurate and inaccurate information. They were also asked what actions they would take if presented with such information (e.g., would they click the link? Would they share it?).

### 4.2. Data Analysis

We employed a rapid analysis approach using template analysis to analyze the workshop transcripts [20,21]. This method involves creating domains for each interview question and developing a template to summarize each transcript by domain [22]. A team of analysts templated the transcripts, with one primary analyst doing an initial templating of the data and a secondary analyst providing a review.

The themes focused on identifying related actionable steps for identifying misinformation. They encompassed various information aspects such as source, messenger, visual appearance, tone, website and URL, content, and trust. Once the data were templated, narratives were extracted based on these dimensions to provide practical guidance for tool development.

### 4.3. Tool Development

Utilizing insights gathered from the workshops, we developed a comprehensive list of steps in Spanish for identifying misinformation. This list was subsequently translated into English to ensure broader accessibility. A designer on the research team created visually engaging elements, tailored to resonate with the target audience. We solicited feedback on the tool from other researchers and participants, ensuring that the final product was visually appealing and effective in helping users identify misinformation. Our collaborative process ensured that the tool was grounded in the real-world experiences of the community members and reflected their perspectives on misinformation identification.

This study was approved by the Institutional Review Board of the University of California San Francisco (IRB protocol #21-34502). Verbal informed consent was obtained from all participants in the formative work and in the workshops in English or Spanish.

## 5. Results

### Formative Interviews and Groups

A total of 30 individual and group interviews were conducted for this study’s formative phase. The impact of misinformation on COVID-19 vaccine uptake emerged as a significant concern among participants in the formative phase of this study. Key informants identified misinformation as a primary reason why many individuals they serve or have talked to refuse or delay getting vaccinated. Group interview participants similarly identified misinformation as a major reason why many of their peers or themselves had not received the vaccine. Some participants who were themselves unvaccinated cited misinformation as the reason for their hesitancy. The prevalence of misinformation online was highlighted, with Spanish speakers disproportionately affected.

Key informant and group interview participants identified several common myths surrounding the COVID vaccine, which were recurrent throughout the formative interviews. These myths were often related to vaccine safety, serious adverse side effects mainly affecting the reproductive system or fertility, conspiracy theories concerning microchips and government control, and doubts about the scientific process, development, and effectiveness of the vaccine. Participants shared personal experiences with these conspiracy theories and expressed concerns about the rapid development and long-term effects on health, particularly for pregnant women and children. The lack of understanding of the approval process and the perception of constantly changing guidelines contributed to vaccine hesitancy, highlighting the need to counter these misconceptions (see Appendix A for more information and findings from the formative interviews). These findings guided the development of the co-design workshops.

## 6. Workshops

We conducted three workshops with three to six participants each in July 2022. All participants were women, self-identified as Hispanic or Latinas, and members of MUA. Most participants were in the age range of 45–54 years, with a diverse range of ages represented (35 to 75 years). All participants spoke Spanish, with the primary language of the two participants in the last workshops being Mam, a Mayan language from Guatemala. Regarding COVID-19 vaccination status, one participant had received one dose, ten were fully vaccinated, and one was unvaccinated.

## 7. Features to Consider for a Co-Designed Toolkit to Counter Misinformation

### 7.1. Source

Participants stressed the need to verify the reliability of sources, including their origins and the destinations of embedded links, noting that even medical professionals can spread misinformation. One participant noted, “when [the link] it is more secure, it always starts with https. And it doesn’t just send you to an unrelated link” (WS1, P5). They highlighted the importance of thoroughly investigating information, especially when it involves significant health decisions for themselves or their families, and recommended seeking input from multiple sources and comparing them rather than relying on a single post.

When presented with two Facebook posts, participants expressed skepticism about Dr. Mercola’s post, with one of them stating, “For me it is garbage or it is not credible because it does not give you access to that information without you having to give personal information or without you having to put your e-mail address and then they invade with advertisements” (WS1, P5). Some participants saw government-related sources as unbiased and trusted, contrasting those to other sources that clearly were profit-based. The Centers for Disease Control (CDC), for example, was viewed as a trustworthy source with free access to information: “To me, the CDC is better, I believe it is the most trusted source. It is giving us all the information. It’s updating us day by day. And it’s giving us a link to keep us informed, it doesn’t say, subscribe or pay” (WS2, P2).

### 7.2. Messenger

Participants stated that it is also important to consider who shared the piece of information with them or the messenger. They reported receiving information primarily through Facebook and WhatsApp and noted that trust in the messenger was a key factor in determining their own level of trust in that information. One participant said she would be more likely to trust COVID-19 information if it was shared by someone she knew and trusted. Another participant added that people can have a strong influence on others, including through fearmongering: “For me it does influence a lot, because even a very close friend tells you: “No, look, it’s because of this and that, and I think that sometimes they do have an influence on you. They also influence you with fear. ‘If you go out, it’s going to happen to you and it’s going to hit you.’” (WS1, P4) The trustworthiness of individuals within one’s social network plays a vital role in shaping their perception and acceptance of shared information, emphasizing the importance of considering both the source and the messenger.

### 7.3. Visual Characteristics

Participants also highlighted the importance of visual presentation when assessing the trustworthiness of COVID-19 information. They recognized that images could have a significant impact on their perception of the information being conveyed. Several participants noted that a photo of a syringe used in Dr. Mercola’s post was aggressive and fear-inducing. However, participants acknowledged that fear-based messaging could have mixed effects on their level of trust. One participant pointed out that images can be particularly influential for illiterate individuals who rely on visual cues to understand the content, stating: “From a visual point of view, the photograph they put up looks rather cruel, because it is like an attack with a syringe… the image that stays with you is ‘Oh, they want to attack us with the vaccine’. They want to manipulate my brain in terms of my image that I’m seeing” (WS1, P2).

### 7.4. Trust in CBOs

The role MUA had in providing them with COVID-related information they could trust was a common theme among participants from the three workshops. As a participant mentioned, belonging to an organization does not only help them be informed but allows them to share with and support others. “And even more so if they don’t have anyone who belongs to an organization, where they are being updated on many things. Because belonging to an organization helps us a lot to be able to help other people. All the information that I receive there, in Mujeres, I am always sharing with the community.” (WS2, P1) Participants expressed trust in community-based organizations, such as MUA, that regularly provided them with COVID-19 vaccine information. The same participant stated, “we trust what MUA gives us, I trust because when they—on Mondays we have the meeting where experts come and give us talks. Every Monday. The people who have come work in hospitals. A doctor has come, a nurse has come, they are people who are informed” (WS2, P1). CBOs played a vital role in bridging the gap between public health and clinical professionals and the communities they serve. Through the trust established with these organizations, they effectively acted as conduits for disseminating accurate, science-based information to the wider community.

### 7.5. Community Characteristics

Participants also discussed how personal and community characteristics can impact trust in vaccine information. They mentioned that people with limited education and exposure to different sources of information may be more vulnerable to misinformation and are more likely to believe everything they read or hear. One participant highlighted that some immigrants may not have had the opportunity to access education or might not be exposed to diverse sources of information, making them more susceptible to believing misinformation: “There are many people, maybe not illiterate, but very humble people who use Facebook and believe everything they say… That’s why people believe anything. They believe anything from anyone” (WS2, P2).

Participants highlighted how fear and shame can hinder individuals in the community from seeking accurate information or asking questions about COVID-19 vaccines. “But, in reality, we are not informed. We don’t know our rights as people. Another thing is that we are afraid to speak up. We are afraid to ask. We are ashamed... That makes the Latino community more intimidated” (WS2, P2). Creating avenues for open dialogue and improving health literacy were central facilitators to establishing trust and encouraging vaccine uptake.

### 7.6. Sources of Misinformation

Participants also cited a wide range of misinformation sources including news media, social media platforms such as Facebook and YouTube, personal doctors, and even religious beliefs. As one participant noted, “Not only people, but the media, YouTube, news and doctors who are [Epidemiologists?]. Yes. They have also come out, many of them, saying that the vaccine is dangerous, and so many things”. (WS1, P3) Participants agreed that in their communities, misinformation was often spread through social media and amplified by personal social networks, as one participant explained: “And, unfortunately, the misinformation we have is precisely because of that, because of what I saw on Facebook and told my comadre and my comadre shared it with my compadre and then shared it with the neighbor, and that’s how misinformation is in our community”. (WS3, P2).

WhatsApp groups were identified as significant sources of both accurate information and misinformation related to COVID-19. One participant revealed their mixed experiences with WhatsApp groups: “I do trust WhatsApp because they send us a lot of information from the organization [MUA]” (WS2, P3). However, participants also acknowledged that there were other WhatsApp groups that disseminated false information.

## 8. Tool

Following the workshops, we developed a comprehensive checklist to assist Spanish speakers in identifying and countering misinformation. The tool was informed by the workshop’s main findings, which highlighted specific themes and strategies that participants deemed crucial in addressing COVID-19 misinformation. This list encompasses ten practical strategies for distinguishing between reliable sources and verifying information on the internet (Figure 1, original Spanish version in Appendix A).

The tool, originally created in Spanish, guides users through a series of simple and effective steps to assess the accuracy of online information. Before finalizing it, it was presented to the women in its original language to ensure that it was comprehensible and matched an appropriate level of literacy. It emphasizes the importance of checking the credibility of sources, cross-referencing with multiple sources, and consulting trusted healthcare providers. It also encourages users to be cautious of alarmist, exaggerated content, or miracle cures, as well as to ensure that links and web addresses appear legitimate. Additionally, the tool highlights the value of staying informed and seeking assistance from trusted community organizations when in doubt. Our tool aims to empower users to navigate online health information with confidence and discernment.

After completing the checklist, we presented it to our collaborators at Mujeres Unidas y Activas (MUA), who then disseminated it among their members. In addition, it was incorporated into a COVID-19 misinformation toolkit, which was created for community-based organizations in the Bay Area. The toolkit is publicly accessible through the UCSF Prevention Research Center’s website at UCSF PRC COVID-19 Vaccine Uptake Project.

## 9. Discussion

The COVID-19 pandemic has disproportionately affected racial and ethnic minorities, including Latino and Spanish-speaking individuals, due to various factors such as health disparities, social determinants, and limited access to accurate health information [15]. Our qualitative study focused on understanding the vaccine misinformation affecting Spanish speakers in the San Francisco Bay Area and developing a user-friendly tool to help them identify and counter it.

Our formative findings revealed that misinformation is a significant concern among participants, with key informants identifying it as a primary reason for vaccine hesitancy. In response, we hosted three workshops with community members to develop a comprehensive checklist to assist Spanish speakers in identifying and countering online COVID-19 vaccine misinformation. The workshops highlighted the importance of evaluating the sources and messengers of information, with participants expressing trust in CBOs and skepticism toward unverified sources or those requiring personal information or payment for access. Participants acknowledged the influence of personal and community characteristics, including low literacy, fear, and shame, on their susceptibility to misinformation and reluctance to seek accurate information. The resulting checklist encompasses practical strategies for distinguishing between reliable sources and verifying information, empowering individuals to seek out and disseminate accurate content online.

Several checklists, guidelines, and initiatives have been developed in other settings to help identify and counter misinformation, with some evidence supporting their effectiveness [23,24,25,26]. For instance, Agley found that briefly viewing a science infographic led to a small aggregate increase in trust in science, potentially reducing the believability of misinformation [11]. Another study evaluated the impact of the WHO misinformation checklist and a modified version they created in Germany and the US, yielding mixed results. While Germans benefited from the tool, Americans did not, suggesting that different populations might require different approaches [27]. To the best of our knowledge, this is the first checklist to address misinformation that is co-designed with the specific target population it is meant to help. Although it shares many commonalities with other existing checklists, such as checking sources and their dates, our tool adds new suggestions that cater to the unique needs of this population, including examining the tone and alarmism of the information, considering potential financial motives, and relying on local CBOs for information.

Participatory design approaches in tool development and public health interventions offer valuable opportunities to gain specific insights into the unique concerns and needs of communities [28]. By involving the target population, co-design workshops ensure that the resulting tools are tailored to the community’s context and address their specific concerns while fostering a sense of ownership and trust in the resulting tools, essential for their successful adoption and use. These methods can identify potential barriers that may hinder the effectiveness of public health interventions, helping to increase inclusivity and accessibility.

It is important to acknowledge the limitations of our study. Our findings may not be generalizable to all Spanish-speaking populations, as the workshops were conducted with a specific group of participants during the COVID-19 pandemic. Additionally, further research is needed to assess the long-term effectiveness of the developed checklist and to adapt it for use in other contexts and populations. Despite these limitations, our study provides valuable insights that can inform the development of future tools and interventions designed to combat online misinformation. Our study underscores the urgent need for effective strategies and tools to combat health misinformation among vulnerable populations, such as Spanish-speaking individuals, at higher risk of being exposed to and affected by misinformation. By increasing digital health literacy, promoting trust in science and health professionals, and investing in culturally appropriate resources and interventions, public health officials can help mitigate the negative consequences of misinformation during public health crises.

## Figures and Tables

**Figure 1 ijerph-21-00294-f001:**
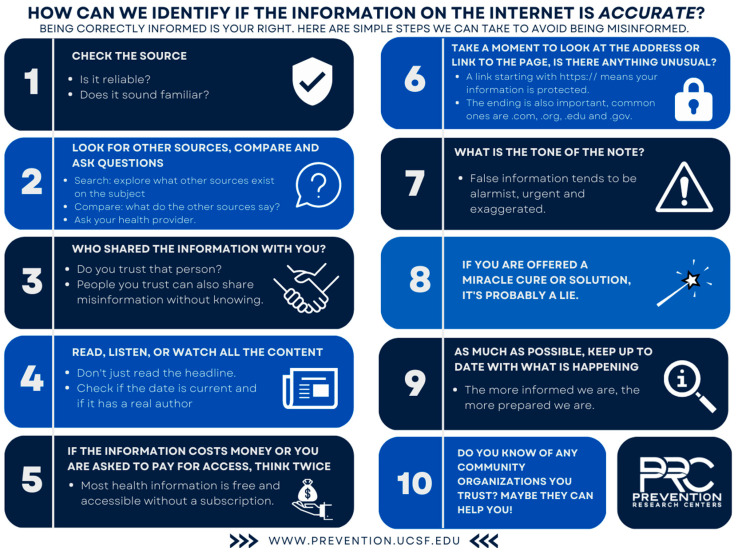
Ten-step list to identify misinformation co-designed with Spanish speakers. Spanish version can be found at: https://prevention.ucsf.edu/about/ucsf-prevention-research-center-prc/ucsf-prc-covid-19-vaccine-uptake-project (access date 20 February2024).

## Data Availability

The data presented in this study are available on request from the corresponding author.

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
