# Peer review of "A Co-Created Tool to Help Counter Health Misinformation for Spanish-Speaking Communities in the San Francisco Bay Area"

_ijerph, 2024, doi:10.3390/ijerph21030294_

Round 1

Reviewer 1 Report

Comments and Suggestions for Authors

Congratulations on an excelent piece of work. If I had to suggest something, I would like to have more information included about the interviewees to have a better idea about the validity of the results. As it is mentioned in the article itself, the Spanish-speaking population object of this study is varied, and knowing more about the informers will help me, as a reader, to know if the sample mirrors this variety. If the informers are experts, knowing their background and expertise will also help with the assessment of their responses.

Author Response

Reviewer 1:

Comment: Congratulations on an excelent piece of work. If I had to suggest something, I would like to have more information included about the interviewees to have a better idea about the validity of the results. As it is mentioned in the article itself, the Spanish-speaking population object of this study is varied, and knowing more about the informers will help me, as a reader, to know if the sample mirrors this variety. If the informers are experts, knowing their background and expertise will also help with the assessment of their responses.

Response:

Thank you for reviewing our paper!

We appreciate the suggestion to provide more detailed information about our interviewees. Understanding the composition of the sample is crucial to assessing the validity of our results, especially given the diversity of the Spanish-speaking population. To this end, we have supplemented the manuscript with additional background information on the organization Mujeres Unidas y Activas (MUA), its members, and the demographics of our workshop participants:

  • We conducted three workshops with three to six participants each in July 2022. All participants were women, self-identified as Hispanic or Latinas, and members of MUA. Most participants were in the age range of 45-54 years, with a diverse range of ages represented (35 to 75 years). All participants spoke Spanish, with the primary language of the two participants in the last workshops being Mam, a Mayan language from Guatemala. Regarding COVID-19 vaccination status, one participant had received one dose, ten were fully vaccinated, and one was unvaccinated.
  • We collaborated with a local CBO, Mujeres Activas y Unidas (MUA), to recruit participants for three 1-hour workshops. Mujeres Unidas y Activas (MUA) is an organization comprised of Latina and Indigenous immigrant women in the San Francisco Bay Area. It is dedicated to empowering both individual and community strengths with the aim of achieving social and economic justice.

Reviewer 2 Report

Comments and Suggestions for Authors

An interesting article based on a well-designed approach. 
That other tools exist is only mentioned in the Discussion. I would expect it to be mentioned under section 8.
Was the ultimate tool presented to the disadvantaged groups that you would like to reach? Wasn't it too cluttered for people with limited literacy?

Comments on the Quality of English Language

Some proofreading required for possible minor mistakes (lines 45, 145, 167; line 155: did they mean https instead of http?). Quotations of what informants said are sometimes in italics, sometimes not.

Author Response

Reviewer 2:

Thank you for reviewing our paper!

Comment: That other tools exist is only mentioned in the Discussion. I would expect it to be mentioned under section 8.

Response: We are grateful for your positive feedback and insightful observations. Regarding the mention of other tools:

  • The reference to other existing tools aimed to acknowledge broader efforts in countering misinformation. We have clarified this in the manuscript to avoid any ambiguity. The revised sentence now reads: "Several checklists, guidelines, and initiatives have been developed in other settings to help identify and counter misinformation, with some evidence supporting their effectiveness [23-26]."

Comment: Was the ultimate tool presented to the disadvantaged groups that you would like to reach? Wasn't it too cluttered for people with limited literacy?

Repsonse: In response to your question about the tool's accessibility, we made sure to include the steps we took for its dissemination:

  • After completing the checklist, we presented it to our collaborators at Mujeres Unidas y Activas (MUA), who then disseminated it among their members. In addition, it was incorporated into a COVID-19 misinformation toolkit, which was created for community-based organizations in the Bay Area. The toolkit is publicly accessible through the UCSF Prevention Research Center's website at UCSF PRC COVID-19 Vaccine Uptake Project.
  • Regarding accesibility: We have not yet empirically tested the tool's effectiveness in less literate populations. However, feedback from our workshop participants indicates that the tool was considered easy to understand. We acknowledge the need for further testing to ensure its suitability for the intended audience.

Comment: Some proofreading required for possible minor mistakes (lines 45, 145, 167; line 155: did they mean https instead of http?). Quotations of what informants said are sometimes in italics, sometimes not.

Response: Lastly, we thank you for pointing out the proofreading oversights:

We have corrected the syntactical errors you identified in lines 45, 145, 167, and the URL reference in line 155. To ensure consistency in the presentation of informant quotations, we have standardized the formatting across the manuscript by using quotation marks exclusively.

45: Low levels of digital health literacy, contribute to the spread of COVID-19-related online misinformation [10,11].

145: Myths surrounding vaccine safety, adverse effects on fertility, conspiracy theories about microchips, and doubts about the vaccine's scientific process and effectiveness were commonly mentioned (see supplemental Table 1 for more information and findings from the formative interviews). These findings guided the development of the co-design workshops.

167: "To me, the CDC is better, I believe it is the most trusted source. It is giving us all the information. Its updating us day by day. And it's giving us a link to keep us informed, it doesn't say, subscribe or pay'" (WS2, P2). 

155: One participant noted, "when [the link] it is more secure, it always starts with https. And it doesn't just send you to an unrelated link" (WS1, P5).